# 3D Gaussian Rendering Can Be Sparser: Efficient Rendering via Learned Fragment Pruning

**Zhifan Ye, Chenxi Wan, Chaojian Li, Jihoon Hong, Sixu Li,**
**Leshu Li, Yongan Zhang, Yingyan (Celine) Lin**
Georgia Institute of Technology
`{zye327, celine.lin}@gatech.edu`

## Abstract

3D Gaussian splatting has recently emerged as a promising technique for novel view synthesis from sparse image sets, yet comes at the cost of requiring millions of 3D Gaussian primitives to reconstruct each 3D scene. This largely limits its application to resource-constrained devices and applications. Despite advances in Gaussian pruning techniques that aim to remove individual 3D Gaussian primitives, the significant reduction in primitives often fails to translate into commensurate increases in rendering speed, impeding efficiency and practical deployment. We identify that this discrepancy arises due to the overlooked impact of fragment count per Gaussian (i.e., the number of pixels each Gaussian is projected onto). To bridge this gap and meet the growing demands for efficient on-device 3D Gaussian rendering, we propose *fragment pruning*, an orthogonal enhancement to existing pruning methods that can significantly accelerate rendering by selectively pruning fragments within each Gaussian. Our pruning framework dynamically optimizes the pruning threshold for each Gaussian, markedly improving rendering speed and quality. Extensive experiments in both static and dynamic scenes validate the effectiveness of our approach. For instance, by integrating our fragment pruning technique with state-of-the-art Gaussian pruning methods, we achieve up to a $1.71\times$ speedup on an edge GPU device, the Jetson Orin NX, and enhance rendering quality by an average of 0.16 PSNR on the Tanks&Temples dataset. Our code is available at `https://github.com/GATECH-EIC/Fragment-Pruning`.

## 1 Introduction

Novel view synthesis, which aims to generate photo-realistic images of a 3D scene from unseen viewpoints given a set of posed multi-view images as input, has been crucial for many virtual reality (VR) and augmented reality (AR) applications [1, 2, 3]. In recent years, Neural Radiance Field [4] (NeRF) and its variants [5, 6, 7] have shown promise in delivering high-quality rendering. However, the volume rendering approach [8] adopted in NeRF requires expensive sampling (e.g., more than 100 times neural network inference per emitted ray [4]) in 3D space, leading to slow rendering speeds. To mitigate this problem, subsequent works have attempted to enhance rendering performance by: (1) storing radiance in explicit 3D representations such as voxel grids [9, 10, 11], octrees [12], and hash grids [13]; and (2) converting pre-trained NeRFs into meshes [14, 15]. However, these methods still rely on the aforementioned costly volume rendering and often lead to increased storage consumption and decreased rendering fidelity. Recently, 3D Gaussian Splatting [16], which adopts a rasterization-based rendering pipeline to avoid the costly point sampling required in volume rendering, has emerged as a promising solution for achieving a superior trade-off between rendering speed and quality. Nevertheless, 3D Gaussian Splatting [16] requires millions of 3D Gaussian primitives to reconstruct a 3D scene (e.g., an average of 3 million per scene in the Mip-NeRF 360 dataset [6]), which still hinders real-time rendering on resource-constrained devices. For instance, as illustrated in

38th Conference on Neural Information Processing Systems (NeurIPS 2024).

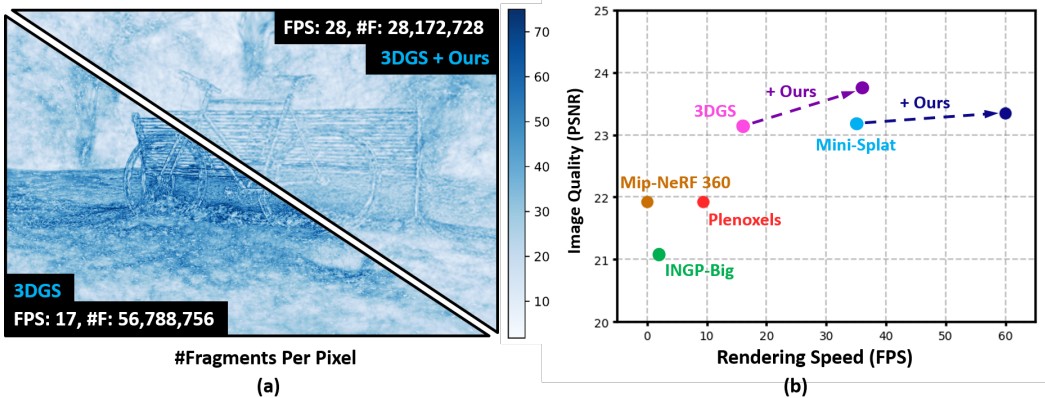

Figure 1: (a) The number of fragments per pixel and the total number of fragments in a bicycle scene from the Mip-NeRF 360 dataset. The bottom left figure corresponds to the vanilla 3D Gaussian Splatting approach (3DGS) [16] while the upper right figure illustrates that of our approach on top of 3DGS. (b) The rendering speed in FPS (Frames Per Second) and image quality in PSNR (Peak Signal-Noise Ratio) of SOTA approaches (including Mip-NeRF 360 [5], INGP-Big [13], Plenoxels [9], 3DGS [16], and Mini-Splatting [18]) and our proposed one.

Sec. 6.3, it achieves only 35 FPS on the Mip-NeRF 360 dataset [6] with a an edge GPU device [17], which is below the required speed for real-time rendering (e.g., 60 FPS).

To further enhance the rendering speed of 3D Gaussian Splatting on edge devices, prior works have proposed pruning individual Gaussian *primitives*. However, as shown in our profiling results in Sec. 4, these prior works do not accelerate the rendering process proportionally to the reduction in the number of primitives, limiting their achievable efficiency. Our further analysis of the rendering process statistics reveals that these existing efforts overlook a critical opportunity, *fragment pruning*, which operates at the fragment, i.e., pixel, granularity. Specifically, we identify that **although these methods significantly reduce the number of Gaussian primitives, they do not proportionally reduce the number of rendered fragments, as the number of fragments per Gaussian increases**. This increase directly impacts computational cost during the bottleneck rasterization stage of 3D Gaussian's rendering pipeline. To the best of our knowledge, we are the first to investigate and develop fragment pruning for 3D Gaussian Splatting.

Based on insights from the aforementioned profiling and analysis, we propose an adaptive fragment pruning framework that learns each Gaussian's fragment pruning threshold as a post-processing step for pre-trained 3D Gaussian Splatting models. Specifically, we use the sigmoid function to approximate the non-differentiable truncation function of a 3D Gaussian projected onto the 2D pixel grid (i.e., the image plane), enabling a differentiable fine-tuning pipeline. This approach enables each Gaussian primitive to learn an independent truncation threshold, thereby reducing the number of pixels it projects onto and, consequently, decreasing the fragment count per Gaussian. This process significantly enhances the rendering efficiency of the truncated Gaussian function. Fig. 1 illustrates the fragment count before and after applying the proposed fragment pruning framework, along with its impact on rendering efficiency and image quality. In summary, our proposed fragment pruning framework makes the following contributions:

- We profile the 3D Gaussian rendering process before and after applying prior Gaussian pruning techniques and derive the following insights: (1) the rasterization stage of the entire rendering pipeline is the bottleneck in terms of runtime, regardless of whether the Gaussian pruning techniques are applied or not; (2) prior works do not reduce the runtime of the bottleneck rasterization stage in proportion to the reduction of the number of Gaussian primitives; (3) this inefficiency arises because prior methods do not proportionally reduce the fragment count, which is a key factor determining the computational cost in the bottleneck rasterization stage.

- Motivated by the insights described above, we propose *fragment pruning*, which prunes 3D Gaussian primitives at the 2D fragment granularity instead of the raw 3D Gaussian primitive. Specifically, we develop an adaptive fragment pruning framework by making the truncation

function for each Gaussian primitive differentiable when projecting it onto the 2D pixel grid. We then fine-tune the pre-trained 3D Gaussian Splatting model using a fully differentiable pipeline, allowing each Gaussian to automatically learn the optimal truncation threshold.

- Extensive experiments and ablation studies on both static and dynamic scenes demonstrate the effectiveness of our proposed fragment pruning. For instance, it is orthogonal to prior Gaussian pruning techniques and can further boost the FPS of state-of-the-art (SOTA) Gaussian pruning works by $1.71 \times$ without decreasing the rendering quality on the Tanks&Temples [19] dataset.

## 2 Related Works

**Novel View Synthesis.** Novel view synthesis aims to render photorealistic images from novel viewpoints, given a set of sparsely sampled images of a specific scene. Toward this goal, early efforts in this field employed classical computer vision techniques such as structure-from-motion [20] and multi-view stereo [21]. More recently, the adoption of deep learning methods has significantly enhanced rendering quality [22, 23, 24, 25]. Specifically, NeRF [4] has emerged as a standout among these deep learning-based novel view synthesis techniques. It models the radiance field of the target scene using implicit neural representations (i.e., multi-layer perceptrons) and achieves unprecedented rendering fidelity. However, NeRF relies on volume rendering [8] with costly sampling of its implicit neural representation along each emitted ray, resulting in slow rendering speeds. To enhance the rendering speeds of NeRF, subsequent research has incorporated explicit geometric structures or more efficient positional encodings to reduce the capacity of the multi-layer perceptron [12, 10, 11, 13], or even eliminated neural networks entirely [9]. Nevertheless, these approaches still adopt the volume rendering pipeline, and the corresponding rendering speeds are limited by the expensive sampling along each emitted ray.

**3D Gaussian Splatting.** Unlike the novel view synthesis works mentioned previously that rely on volume rendering, 3D Gaussian Splatting [16] parametrizes a scene using a set of 3D Gaussian primitives and utilizes a GPU-friendly rasterization-based rendering pipeline. This approach circumvents the need for expensive sampling along each emitted ray. In addition to improvements in rendering speed, the use of continuous and anisotropic Gaussian primitives also enables faster convergence and higher reconstruction fidelity compared to the aforementioned NeRF-based methods. Consequently, this approach has inspired numerous follow-up works that extend its application to dynamic scene reconstruction [26, 27, 28] and 3D content generation [29, 30], demonstrating superior performance across various domains.

**3D Gaussian Splatting Pruning.** Despite the impressive rendering quality and efficiency achieved by 3D Gaussian Splatting [16], high redundancy in the reconstructed 3D Gaussian models adversely affects rendering speeds, as identified by [31, 18]. To further boost rendering speeds by compressing this identified redundancy, various techniques have been proposed to prune the redundant Gaussian primitives, drawing parallels to prior deep neural network pruning efforts [32, 33]. For example, Compact 3D Gaussian [31] learns a binary mask to remove unnecessary Gaussians. LightGaussian [34] introduces an importance measure to eliminate insignificant Gaussians below a certain threshold. Mini-Splatting [18] uses importance-based sampling rather than a strict pruning threshold to mitigate excessive culling in local regions. Similarly, [35] proposes a redundancy-based sampling method for Gaussian removal. However, all the aforementioned 3D Gaussian Splatting pruning techniques only address pruning at the granularity of individual Gaussian *primitives*, neglecting the impact of *the number of fragments*, produced after projecting 3D Gaussians onto the 2D image plane, on rendering speeds. In contrast, our work explores an orthogonal direction at the granularity of fragments, significantly enhancing the trade-offs between rendering speeds and quality in both static and dynamic scenes, as validated in Sec. 6.

**NeRF to Mesh Convertion** In addition to 3D Gaussians, polygon meshes also offer an efficient 3D scene representation with a rendering pipeline that is well-optimized on commercial devices. Previous works [14, 15, 36] have proposed converting pre-trained NeRFs into opaque polygon meshes to enhance rendering speed in novel view synthesis tasks. After conversion, a pixel's color depends solely on the closest projected polygon, which reduces per-pixel computational costs but makes it challenging to faithfully reconstruct complex geometry and translucent objects. Consequently, the

converted mesh often struggles to achieve high rendering quality in real-world scenes. For instance, on the Mip-NeRF 360 dataset [6], the prior works on converting NeRF to mesh [14, 15, 36] achieved PSNR scores of $21.95 \sim 23.59$, lower than the vanilla NeRF [4], which reached a PSNR of 23.85 [6]. In contrast, our approach, which adaptively reduces computational cost per pixel (i.e., fragment count per pixel, as shown in Fig. 1 (a)) based on learned truncation thresholds, preserves the rendering quality of pre-trained 3D Gaussian models.

## 3 Preliminary

### 3.1 Gaussian Primitives of 3D Gaussian Splatting

3D Gaussian Splatting [16] represents a scene with a collection of anisotropic Gaussian primitives. Each primitive is parameterized by a 3D Gaussian function $G(\mathbf{x})$, with a covariance matrix $\mathbf{\Sigma} \in \mathbb{R}^{3 \times 3}$, centered at a mean value (center point) $\mu \in \mathbb{R}^3$:

$$G(\mathbf{x}) = e^{-\frac{1}{2}(\mathbf{x}-\mu)^T \Sigma^{-1}(\mathbf{x}-\mu)} \tag{1}$$

The covariance matrix $\mathbf{\Sigma}$ can be further decomposed into a rotation matrix $\mathbf{R}$ and a scaling matrix $\mathbf{S}$: $\mathbf{\Sigma} = \mathbf{R}\mathbf{S}\mathbf{S}^T\mathbf{R}^T$. Besides storing the aforementioned geometry information, each Gaussian is also assigned a learnable opacity factor $o$ and a set of spherical harmonics (SH) coefficients for view-dependent colors, following previous practices [9, 13].

### 3.2 Rendering Pipeline of 3D Gaussian Splatting

Given the aforementioned 3D Gaussian primitives and a specific viewpoint (camera), the rendering pipeline for 3D Gaussian Splatting can be divided into three stages:

**Projection** First, each 3D Gaussian primitive is projected onto the 2D image plane to be rendered by taking the marginal distribution of the 3D Gaussian function. This stage results in a set of 2D Gaussian primitives [37]. Specifically, using a viewing transformation $\mathbf{W}$ and a Jacobian matrix $\mathbf{J}$, which approximates the local projective transformation with an affine mapping, the covariance matrix $\mathbf{\Sigma}'$ of the 2D primitive can be derived as follows:

$$\mathbf{\Sigma}' = \mathbf{J}\mathbf{W}\mathbf{\Sigma}\mathbf{W}^T\mathbf{J}^T. \tag{2}$$

**Sorting** Second, all Gaussian primitives are sorted by their depth values $d$, i.e., the distance from their centers to the camera origin. As a result, the Gaussian primitives are ranked in order from nearest to farthest relative to the viewpoint:

$$d_1 < d_2 < d_3 < ... < d_{\#G}, \tag{3}$$

where $\#G$ is the total number of Gaussian primitives.

**Rasterization** Finally, for each pixel on the 2D image plane to be rendered, the color and opacity of the sorted 2D Gaussian primitives are rasterized and accumulated through $\alpha$-blending:

$$\mathbf{C}(\mathbf{p}) = \sum_{i=1}^{\#G} \mathbf{c}_i \alpha_i(\mathbf{p}) \prod_{j=1}^{i-1}(1 - \alpha_i(\mathbf{p})), \tag{4}$$

where $\mathbf{p}$ is the pixel center and the opacity value $\alpha_i(\mathbf{p})$ is determined by sampling a weighted 2D Gaussian function at location $\mathbf{p}$:

$$\alpha_i(\mathbf{p}) = o_i e^{-\frac{1}{2}(\mathbf{p}-\mu_\mathbf{i})^T \Sigma_i'^{-1}(\mathbf{p}-\mu_i)}. \tag{5}$$

In practice, it is computationally infeasible to sample Eq. 5 at all pixels for all Gaussian primitives, which involves multiple vector-matrix multiplications. Therefore, 3D Gaussian Splatting [16] truncates each 2D Gaussian function at a predetermined $\alpha$ threshold $T = \frac{1}{255}$ and ignores the impact of 2D Gaussian primitives on the pixels falling outside the truncation range. This approach essentially replaces opacity value $\alpha_i$ with $\alpha_i'$ in Eq. 4:

$$\alpha_i'(\mathbf{p}) = \alpha_i(\mathbf{p})\mathbb{I}\{\alpha_i(\mathbf{p}) > T\}, \tag{6}$$

where $\mathbb{I}$ is an indicator function that equals 1 if and only if the condition is met, and equals 0 otherwise.

Table 1: Per-scene rendering time breakdown for vanilla [16] and pruned [18] 3D Gaussian Splatting. Rendering times are measured in milliseconds, based on the first camera pose in the test set. The latency at each stage is stable for this fixed pose, with fluctuations below 0.02 milliseconds. Specifically, #G and #F represent the number of Gaussians and the number of fragments for the corresponding scene.

| Scene | Pruned | Projection | Sorting | Rasterization | #G | #F | #F / #G |
|-------|--------|------------|---------|---------------|-----|-----|---------|
| Bicycle | ✗ | 7.64 | 6.61 | 42.57 | 5,991,553 | 56,788,756 | 9.5 |
|         | ✓ | 1.32 | 0.90 | 18.00 | 533,288 | 34,633,815 | 64.9 |
| Garden | ✗ | 9.92 | 8.08 | 45.50 | 5,742,553 | 58,067,106 | 10.1 |
|        | ✓ | 1.58 | 1.45 | 20.69 | 575,510 | 33,342,349 | 57.9 |
| Room | ✗ | 2.37 | 2.43 | 47.20 | 1,535,973 | 83,986,935 | 54.7 |
|      | ✓ | 1.10 | 0.87 | 32.39 | 393,991 | 76,513,014 | 194.2 |
| Kitchen | ✗ | 6.28 | 4.71 | 59.44 | 1,809,084 | 99,977,477 | 55.3 |
|         | ✓ | 1.41 | 1.01 | 36.88 | 432,212 | 82,708,870 | 191.4 |

## 4 Profiling and Analysis of 3D Gaussian Rendering Pipeline

To understand the bottlenecks of the aforementioned 3D Gaussian Splatting rendering pipeline, we perform profiling on an OpenGL-accelerated Gaussian Splatting renderer [38] using the Jetson Orin NX edge GPU system [17]. Specifically, in the profiling experiments, we adopt two outdoor scenes, "Garden" and "Bicycle", and two indoor scenes, "Room" and "Kitchen", from the Mip-NeRF 360 dataset [6]. To quantify the impact of prior 3D Gaussian Splatting pruning on rendering speeds, we profile both the vanilla 3D Gaussian Splatting and Mini-Splatting [18], a SOTA method that reduces Gaussian primitives by 85.4% while achieving higher rendering quality compared to the vanilla one.

### 4.1 Understanding the Efficiency Bottleneck in the Rendering Pipeline

As summarized in Tab. 1, the **Rasterization** stage is the major bottleneck in rendering speed for both the vanilla and pruned 3D Gaussian Splatting. Specifically, this stage accounts for 71.66% to 94.25% of the overall rendering runtime.

The runtime is dominated by the **Rasterization** stage because: (1) for the **Projection** and **Sorting** stages, which perform per Gaussian projection and sort all the Gaussians based on depth order, respectively, the computational complexity is determined by the number of Gaussians ($\#G$) to be rendered; (2) in contrast, the **Rasterization** stage's computational time is dominated by per fragment computation (i.e., Eq. 4), whose complexity is mainly determined by the number of fragments (i.e., the sum of pixels covered by truncated 2D Gaussian primitives):

$$\#F = \sum_{i=1}^{\#G} \sum_{\mathbf{p} \in \mathcal{P}} \mathbb{I}\{\alpha_i(\mathbf{p}) > T\}. \tag{7}$$

Based on the summary in Tab. 1, the number of fragments ($\#F$) is one to two orders of magnitude larger than the number of Gaussians ($\#G$) under all experimental settings, which results in higher computational complexity and longer rendering latency. This is why **Rasterization** is the primary bottleneck among the three stages.

### 4.2 Understanding the Effect of Pruning 3D Gaussian Primitives

As summarized in Tab. 1, we observe that pruning 3D Gaussian primitives with SOTA pruning techniques [18] results in varying degrees of speedup across the three stages of the rendering pipeline. Specifically, when the number of Gaussians ($\#G$) is reduced by $7.79\times$ by [18], the **Projection** and **Sorting** stages experience speedups of $4.85\times$ and $5.16\times$, respectively, while the **Rasterization** stage only gains a speedup of $1.81\times$. This implies that prior 3D Gaussian pruning efforts do not proportionally reduce the runtime of the bottleneck stage relative to the reduction in the number of Gaussian primitives.

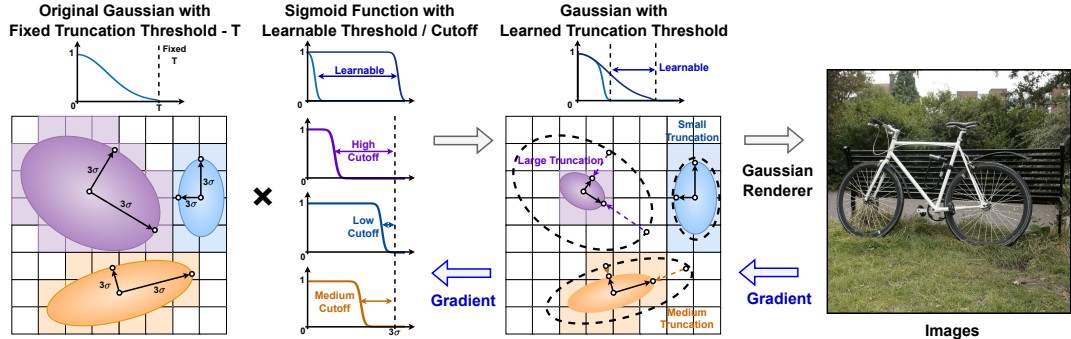

Figure 2: The overall pipeline of our proposed fragment pruning framework. We apply a learnable truncation threshold for each Gaussian to reduce the number of pixels (i.e., fragments) covered by each Gaussian primitive.

Leveraging the aforementioned analysis on computational complexity across the three different stages, and considering the number of Gaussian primitives ($\#G$) and fragments ($\#F$) before and after pruning as summarized in Tab. 1, we can conclude the following: The less-than-expected reduction in runtime is primarily because the number of fragments ($\#F$) only decreases by $1.13\times$ after pruning, which is caused by an increased average number of fragments per Gaussian primitive ($\#F$ / $\#G$). This leads to a limited reduction in the computational complexity at the bottleneck **Rasterization** stage.

## 5 The Proposed *Fragment Pruning* Framework

Motivated by the profiling and analysis in Sec. 4, which reveals that prior 3D Gaussian Splatting pruning efforts overlooked the impact of the number of fragments, we propose *fragment pruning*. This approach performs pruning at the granularity of fragments rather than at the raw 3D Gaussian primitive level, as shown in Fig. 2. Specifically, we detail the pruning mechanism via adaptive Gaussian truncation in Sec. 5.1 and describe the differentiable learning pipeline for determining the optimal truncation threshold in Sec. 5.2.

### 5.1 Fragment Pruning through Adaptive Gaussian Truncation

Based on the insights summarized in Sec. 4, we propose pruning the number of fragments covered by the $i$-th 3D Gaussian primitive, i.e.,

$$\#F_i = \sum_{\mathbf{p} \in \mathcal{P}} \mathbb{I}\{\alpha_i(\mathbf{p}) > T\}, \tag{8}$$

by increasing the threshold $T$ as compared to the default value used in the vanilla 3D Gaussian Splatting. Specifically, to accommodate the size variations of different 3D Gaussian primitives, we propose learning a per Gaussian truncation threshold $T_i$, rather than adopting a global threshold $T$. Thus, the opacity value for the $i$-th 3D Gaussian primitive, as shown in Eq. 6, can be formulated as follows:

$$\alpha_i'(\mathbf{p}) = \alpha_i(\mathbf{p})\mathbb{I}\{E_i(\mathbf{p}) > T_i\}. \tag{9}$$

It is worth noting that to avoid the computational workload associated with calculating exponentials for Eq. 5, we truncate Gaussian primitives based on the exponent term in Eq. 5 (denoted as $E_i$)[1], instead of directly on $\alpha_i$.

### 5.2 Differentiable Learning of Truncation Thresholds

To locate the optimal truncation threshold $T_i$ for each Gaussian primitive, our proposed fragment pruning framework starts with a pre-trained 3D Gaussian model and fine-tunes it using a fully differentiable pipeline.

---

[1]Truncating based on $E_i$ is mathematically equivalent to truncating based on $\alpha_i$. The equivalent truncation threshold for $\alpha_i$ is $o_i e^{T_i}$.

**During fine-tuning**, the key barrier to building a fully differentiable pipeline is the non-differentiable indicator function $\mathbb{I}$. Thus, we propose approximating $\mathbb{I}$ with the differentiable *Sigmoid* function:

$$\mathbb{I}\{E_i(\mathbf{x}) > T_i\} \approx Sigmoid(\frac{E_i(\mathbf{x}) - T_i}{t}), \tag{10}$$

where $t$ is a positive constant. We set $T_i$ and $o_i$ as trainable during fine-tuning and freeze all other parameters, including $R$, $S$, and the SH coefficients. This is because we empirically find that fine-tuning the entire model can lead to overfitting issues.

**After fine-tuning**, we discard the approximation and directly adopt Eq. 9 for rendering with a reduced number of fragments. Therefore, there is no need to compute the $Sigmoid$ function during rendering, and no extra rendering time is incurred due to the approximation in Eq. 10.

# 6 Experiments

## 6.1 Experiment Setup

**Datasets.** We evaluate our proposed techniques on both static and dynamic scenes. For static scenes, we adopt the five outdoor scenes and four indoor scenes from the **Mip-NeRF 360** dataset [6], two scenes ("Train" and "Truck") from the **Tanks&Temples** dataset [19] and two scenes ("DrJohnson" and "Playroom") from the **Deep Blending** dataset [22]. For dynamic scenes, we select the **Plenoptic Video Dataset** [39], which is composed of six real-world video sequences.

**Devices.** To validate the effectiveness of the proposed approach, we benchmark the rendering speed of our method and the baselines on a consumer hardware device, Nvidia's edge GPU, the Jetson Orin NX [17]. The rendering resolution for all scenes is fixed at 1080P (1920 × 1080) to match the minimal requirements of AR/VR applications [40, 41, 42].

## 6.2 Implementation Details

**Pre-training and Initialization.** We use the official implementations and the default hyperparameters of 3D Gaussian Splatting [16], Mini-Splatting [16], and 4D Gaussian Splatting [28] to obtain pre-trained models. Before fine-tuning, we initialize the truncation threshold $T_i$ for each Gaussian based on its opacity:

$$T_i = log(\frac{1}{64 \cdot o_i}), \tag{11}$$

so that after initialization, the effective threshold on $\alpha_i$ is $\frac{1}{64}$.

**Fine-tuning on Static Scenes.** To validate the robustness of the proposed techniques, we use the same set of hyperparameters to fine-tune all scenes across different datasets. Specifically, we fine-tune each scene for 5,000 epochs, utilizing a batch size of 1. In particular, we adopt the Adam optimizer with a learning rate of 0.01, $\beta_1 = 0.9$, and $\beta_2 = 0.99$ during the fine-tuning process. We adopt the same L1 Loss and SSIM Loss as the pre-training process [16].

**Fine-tuning on Dynamic Scenes.** In dynamic scenes, we maintain the same hyperparameters, optimization strategies and loss functions as those used in static scenes. However, we adjust our training batch size to 4, adhering to the default batch size as specified in the 4D Gaussian Splatting training [28].

**Renderer Implementation.** To fully leverage the hardware-accelerated graphics pipeline of consumer GPUs, we conducted performance benchmarks for vanilla 3D Gaussian Splatting [16] and Mini-Splatting [18] using an OpenGL-based renderer [38]. Additionally, we have enhanced this renderer to incorporate our adaptive Gaussian truncation technique and to facilitate the rendering of dynamic scenes using 4D Gaussian Splatting [28].

## 6.3 Quantative Results on Static Scenes

Tab. 2 presents a quantitative comparison of the proposed approach and baseline methods on the static scenes. We report the average Peak Signal-to-Noise Ratio (PSNR), Structural Similarity Index

Table 2: Quantitative comparison of our method and previous works on the Plenoptic Video Dataset. Frame rates (FPS) of our method and 4DGS [28] are measured on Jetson Orin NX [17]. *: Frame rates capped at 60 FPS. In the Mip-NeRF 360 dataset, Mini Splatting + Ours achieved 60 FPS on 5 out of the 9 scenes. In the Tanks&Temples and Deep Blending dataset, all scenes achieved at 60 FPS.

| Dataset | Mip-NeRF 360 | | | | Tanks&Temples | | | | Deep Blending | | | |
|---|---|---|---|---|---|---|---|---|---|---|---|---|
| Method \| Metric | PSNR↑ | SSIM↑ | LPIPS↓ | FPS↑ | PSNR↑ | SSIM↑ | LPIPS↓ | FPS↑ | PSNR↑ | SSIM↑ | LPIPS↓ | FPS↑ |
| Plenoxels [9] | 23.08 | 0.626 | 0.436 | - | 21.08 | 0.719 | 0.379 | - | 23.06 | 0.795 | 0.510 | - |
| INGP-Big [13] | 25.59 | 0.699 | 0.331 | - | 21.92 | 0.745 | 0.305 | - | 24.96 | 0.817 | 0.390 | - |
| Mip-NeRF 360 [6] | 27.69 | 0.792 | 0.331 | - | 21.92 | 0.745 | 0.305 | - | 24.96 | 0.817 | 0.390 | - |
| 3D Gaussian [16] | 27.21 | 0.815 | 0.214 | 20 | 23.14 | 0.841 | 0.183 | 16 | 29.41 | 0.903 | 0.243 | 18 |
| 3D Gaussian [16]* | 27.44 | 0.813 | 0.218 | 20 | 23.71 | 0.848 | 0.177 | 16 | 29.55 | 0.904 | 0.244 | 18 |
| + Ours | 27.48 | 0.812 | 0.209 | 37 | 23.75 | 0.845 | 0.176 | 36 | 29.60 | 0.899 | 0.243 | 44 |
| Mini-Splatting [18] | 27.34 | 0.822 | 0.217 | 35 | 23.18 | 0.835 | 0.202 | 35 | 29.98 | 0.908 | 0.253 | 39 |
| **+ Ours** | **27.38** | **0.822** | **0.209** | **54*** | **23.34** | **0.835** | **0.200** | **60*** | **30.04** | **0.905** | **0.252** | **60*** |

Measure (SSIM), Learned Perceptual Image Patch Similarity (LPIPS), and Frames per Second (FPS) on the Mip-NeRF-360, Tanks&Temples, and Deep Blending datasets.

Our proposed fragment pruning *consistently enhances rendering speed while maintaining or improving rendering quality* across various datasets. When applied to pre-trained 3D Gaussian Splatting models, this technique significantly increases FPS, with gains of +17, +20, and +26 across three different datasets, effectively more than doubling the rendering frame rates. Concurrently, it also boosts the PSNR by +0.04, +0.04, and +0.05, respectively, across the three datasets.

Furthermore, the data presented in Tab. 2 illustrate that *the proposed fragment pruning approach outperforms the SOTA Gaussian primitive pruning technique*, i.e., Mini-Splatting [18], in terms of the rendering speed vs. quality trade-offs. Across all three real-world datasets, the proposed fragment pruning consistently achieves higher FPS than Mini-Splatting, while maintaining superior or comparable rendering quality. For example, on the Tanks&Temples dataset [19], applying our fragment pruning technique to a pre-trained 3D Gaussian Splatting model achieves a 0.57 PSNR improvement and a 1 FPS increase over Mini-Splatting [18].

Most importantly, our experimental findings, as shown in Tab. 2, demonstrate that *the proposed fragment pruning technique complements the SOTA Gaussian primitive pruning approach* [18]. By integrating our fragment pruning method with Mini-Splatting [18], we achieved significant enhancements in rendering efficiency. Specifically, the application of fragment pruning in conjunction with Mini-Splatting resulted in increases in rendering speeds by +19, +25, and +21 FPS, equating to improvements of +54%, +71%, and +54% on the Mip-NeRF-360, Tanks&Temples, and Deep Blending datasets, respectively, while maintaining or even enhancing rendering quality. Notably, on the Tanks&Temples and Deep Blending datasets, the combined use of fragment pruning and Gaussian primitive pruning achieved the maximum screen refresh rate of 60 FPS, underscoring the efficacy of these combined techniques in practical scenarios.

## 6.4   Quantative Results on Dynamic Scenes

Tab. 3 summarizes the quantitative improvements achieved by integrating fragment pruning into 4D Gaussian Splatting [27]. In real-world dynamic scenes, the implementation of this pruning method

Table 3: Quantitative comparison of our method and previous works on the Plenoptic Video Dataset. [1]: numbers from 4DGS paper; [2]: FPS only measured on four scenes, i.e. without Coffeer Martini and Flame Salmon, which run out of memory on Jetson NX.

| Method | PSNR↑ | SSIM↑ | LPIPS↓ | FPS↑ |
|---|---|---|---|---|
| K-Planes-explicit [43][1] | 30.88 | 0.020 | - | 0.23 |
| K-Planes-hybrid [43][1] | 31.63 | 0.018 | - | - |
| MixVoxels-L [44][1] | 30.80 | 0.020 | 0.126 | 16.7 |
| NeRFPlayer [45][1] | 30.69 | 0.035 | 0.111 | 0.045 |
| HexPlane [46][1] | 31.70 | 0.014 | 0.075 | 0.56 |
| HyperReel [47][1] | 31.10 | 0.037 | 0.096 | 2.00 |
| 4DGS [48][2] | 32.01 | 0.030 | 0.055 | 14 |
| **+ Ours**[2] | **32.04** | **0.030** | **0.053** | **22** |

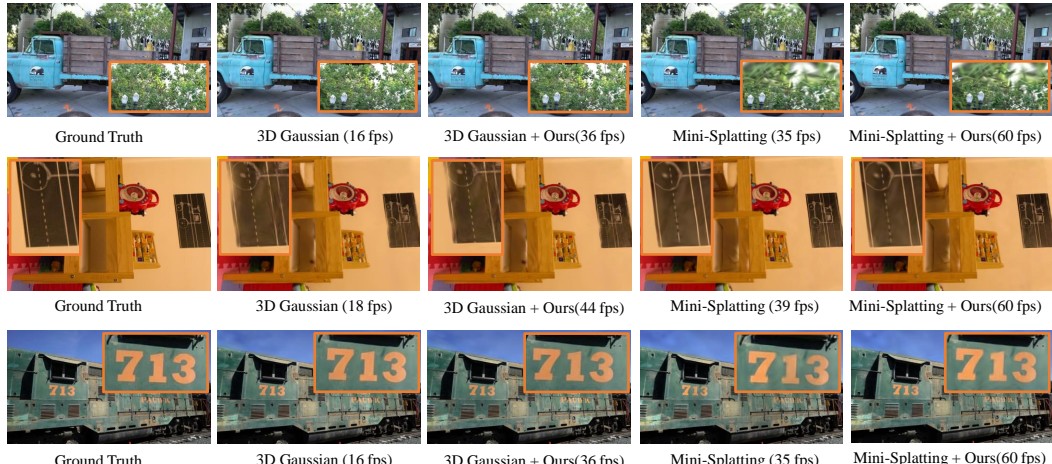

| Ground Truth | 3D Gaussian (16 fps) | 3D Gaussian + Ours(36 fps) | Mini-Splatting (35 fps) | Mini-Splatting + Ours(60 fps) |

| Ground Truth | 3D Gaussian (18 fps) | 3D Gaussian + Ours(44 fps) | Mini-Splatting (39 fps) | Mini-Splatting + Ours(60 fps) |

| Ground Truth | 3D Gaussian (16 fps) | 3D Gaussian + Ours(36 fps) | Mini-Splatting (35 fps) | Mini-Splatting + Ours(60 fps) |

Figure 3: Qualitative comparison of 3D Gaussian Splatting and different Gaussian pruning methods. Zoom in for a better view.

results in an average increase in rendering speed of +8 FPS, maintaining the same quality of rendering. This enhancement is facilitated by the direct application of hyperparameters optimized for static scenes, demonstrating the robustness and versatility of the fragment pruning strategy.

### 6.5 Qualitative Results

Fig. 3 qualitatively compares the rendering output from vanilla 3D Gaussian Splatting [16], 3D Gaussian Splatting with the proposed fragment pruning approach, Mini-Splatting [18] and Mini-Splatting with the proposed fragment pruning approach. Based on the first row of Fig. 3, we can observe that prior Gaussian primitive pruning approaches like Mini-Splatting result in severe artifacts in some distant regions (e.g., the leaves). In contrast, the proposed fragment pruning approach better preserves the rendering quality in such areas while achieving similar or better rendering efficiency. From the second row of Fig. 3, we can conclude that although Mini-Splatting removes the blurry reconstruction, it can results in new artifacts on reconstructing the dotted line. Additionally, as shown in the last column of the second row, learning per Gaussian truncation threshold in our proposed fragment pruning can partially recover the rendering quality. Similarly, on the last row of Fig. 3, Mini-Splatting can result in a drop in rendering quality (a less clear number "1" as compared to the vanilla 3D Gaussian Splatting), while our proposed fragment pruning can recover the reconstruction quality. More qualitative results are presented in the appendix.

## 7 Discussion

### 7.1 Limitation

One limitation of this work is the additional fine-tuning time required, since we designed fragment pruning as a post-training fine-tuning technique. As shown in Tab. 4, the fine-tuning stage incurs an overhead of 16% -

Table 4: Comparison of pre-training and fine-tuning times on an A5000 GPU (in minutes).

| Dataset | On 3D Gaussian [16] | | On Mini-Splatting [18] | |
|---|---|---|---|---|
| | Pre-training | Fine-tuning | Pre-training | Fine-tuning |
| Mip-NeRF 360 [6] | 45.2 | 7.7 | 17.2 | 3.4 |
| Tanks&Temples [19] | 27.2 | 4.4 | 11.4 | 2.2 |
| Deep Blending [22] | 37.4 | 7.1 | 16.2 | 3.4 |

21% relative to the pre-training time. Integrating threshold learning into the original pre-training process is a potential solution, which we leave for future work.

### 7.2 Future Works

There is a plethora of downstream tasks that leverage the benefits of 3D Gaussian Splatting models including the generation of 3D contents and dynamic scene construction. In order to generate 3D objects from a single image of text prompt, DreamGaussian [29] and GSGen [49] combine score distillation sampling with 3D Gaussian primitives while LGM [50] utilize an asymmetric UNet

architecture to predict the parameters of Gaussian features. On the other hand, multiple works have extended 3D Gaussian Splatting models to incorporate time by introducing time-variant opacity [51] or directly including time as a 4D Gaussian parameter [27], [28] to generate dynamic scenes. Our work has focused on the efficiency of our algorithm on dynamic scene generation using 4D Gaussian features, and we leave the application of fragment pruning to other tasks as future research.

## 8 Conclusion

In this work, we propose a novel fragment pruning algorithm that effectively improves the rendering speed of 3D Gaussian Splatting models. We show through rigorous profiling that the main bottleneck of the rendering process is the rasterization stage whose computational complexity is determined by number of fragments. Building on this finding, we suggest an algorithm that learns the truncation threshold for each Gaussian instead of using a predetermined heuristic threshold, which allows to truncate Gaussians in a more aggressive manner and thereby reduce the number of fragments. As a result, our method has achieved significant rendering speed improvements over the original 3D Gaussian model with negligible deterioration or even an improvement of quality. While our algorithm alone demonstrates superior performance gains over SOTA Gaussian primitive pruning techniques, it can be used in conjunction with SOTA primitive pruning algorithms to further boost the rendering speed of 3D Gaussian models.

## Acknowledgments and Disclosure of Funding

This work is supported by the NSF Computing and Communication Foundations (CCF) program (Award ID: 2312758), and the CoCoSys, one of the seven centers in JUMP 2.0, a Semiconductor Research Corporation (SRC) program sponsored by DARPA. Furthermore, we extend our gratitude towards the reviewers of this paper for their insightful comments and suggestions.

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

# A  More Rendering Output Visualization

Fig. 4 provides more visualization on the rendering output from vanilla 3D Gaussian Splatting [16], 3D Gaussian Splatting + the proposed fragment pruning approach, Mini-Splatting and Mini-Splatting + the proposed fragment pruning approach.

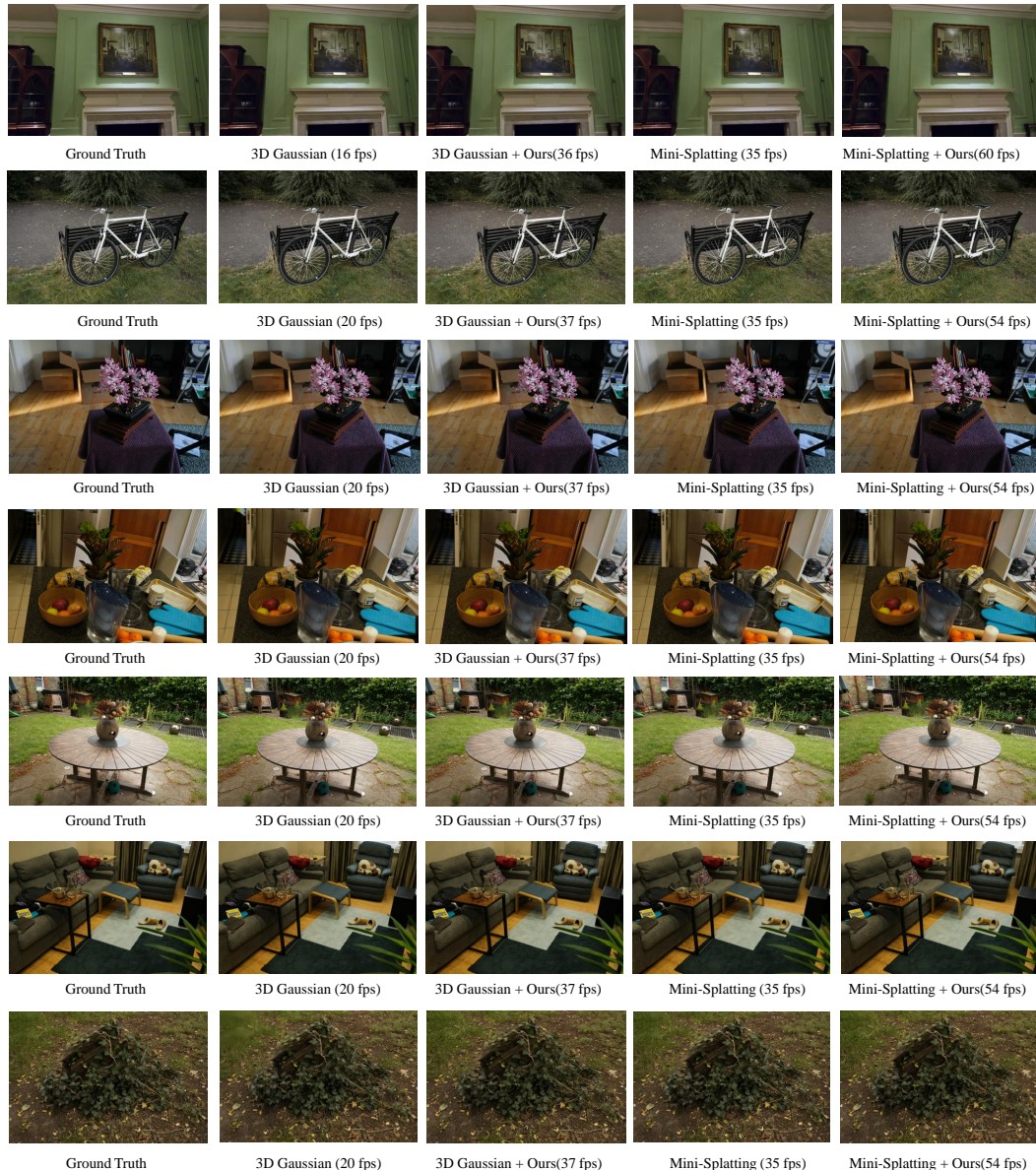

Figure 4: Qualitative comparison between vanilla 3D Gaussian Splatting, state-of-the-art (SOTA) Gaussian primitive pruning work, Mini-Splatting [18], and our proposed Fragment Pruning.

# B  More Fragment Density Visualization

Fig. 5 compares the rendering quality and per-pixel fragment density before and after applying the proposed fragment pruning technique on pre-trained 3D Gaussian models [16]. As observed in column (c), the object borders exhibit the highest fragment density. By reducing the fragment density at these borders, as shown in column (d), our method enhances rendering quality, especially along object edges (illustrated in column (b)).

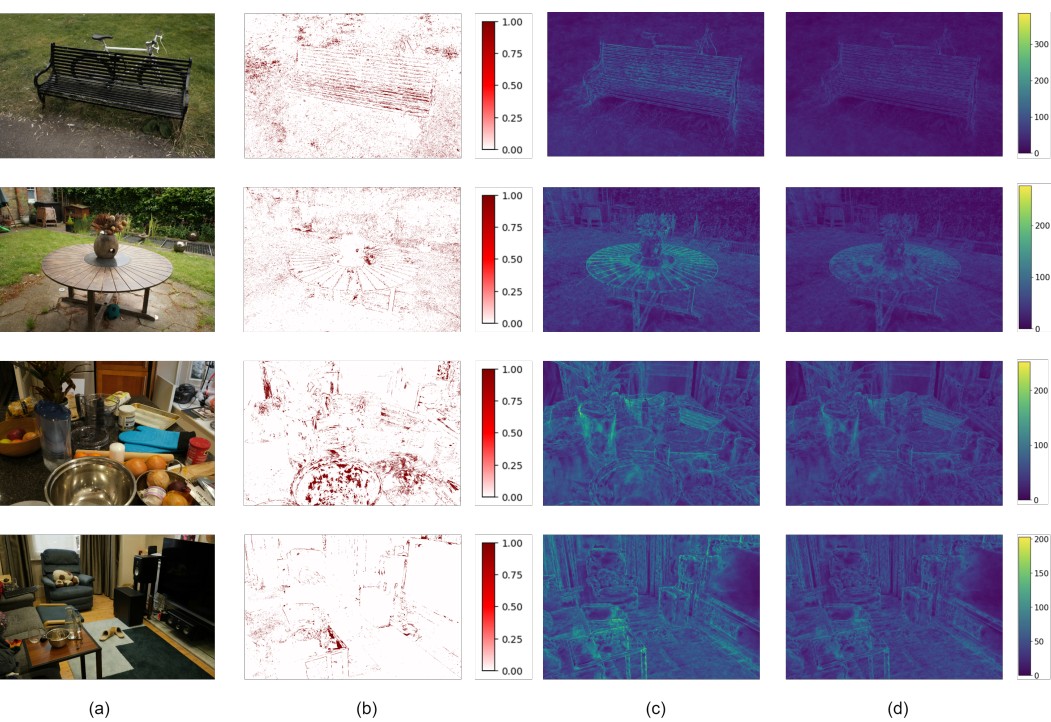

Figure 5: Visualizing of rendering quality improvement and fragment density reduction. (a) Ground truth image. (b) Regions where fragment pruning improves rendering fidelity, measured by a reduction in per-pixel L1 error. (c) Fragment density before fragment pruning. (d) Fragment density after fragment pruning.

