# OpenReview forum: "3D Gaussian Rendering Can Be Sparser: Efficient Rendering via Learned Fragment Pruning"
_NeurIPS.cc/2024/Conference — NeurIPS 2024 poster_

### Official Review · Reviewer_7Aqo · 2024-06-30

**Soundness:** 3
**Presentation:** 2
**Contribution:** 2
**Rating:** 5
**Confidence:** 5

**Summary:**

This manuscript proposes a differentiable pruning strategy on 3D Gaussian splatting by performing the pruning stage after full optimization and continuing the optimization of the Gaussians with additional parameters that determine when to prune the Gaussians when they splat to 2D ( 2D pruning instead of the normal 3D pruning [19]). The proposed method gains marginal rendering speed compared to 3DGS on standard novel view synthesis benchmarks.

**Strengths:**

- Pruning 3D Gaussians splatting is a relevant topic of research with several works trying to address it.
- The code is provided

**Weaknesses:**

- The work suffers limited novelty. The proposed profiling, adding fragment pruning compares to the naive Gaussian pruning [19], and the resulting marginal improvement seems like a great engineering work that is not most suitable for a Neurips paper. The scope is limited such that there is no technical insight that can be used elsewhere in other 3D computer vision research areas based on the discussions provided in this paper. Next year when people stop using 3D Gaussian Splatting and start using something else ( as they did with NeRFs), how will this research be relevant?
- The pipeline requires **TWO** full scene optimizations, one with the native 3DGS and the other with adding the differentibe fragment pruning. This added cost is not properly reflected in the results in Table 1 and Table 2. , which is not very scientific. The gained rendering speed is also not substantial enough to justify the cost in my opinion.
- Why is there a gain in PSNR after pruning? Does this seem counterintuitive ?! The naive optimization can just easily reduce the opacity to 0 and get the same results instead of pruning it.

**Questions:**

inital Rating week reject

Reason: The work provides an alternative Gaussian Splatting pruning based on 2D fragments instead of 3D pruning [19], the proposed alternative is a major engineering feature that lacks technical insight, with marginal rendering speed improvement compared to the slow optimization ( twice ). The scope of the work is limited to the current 3DGS and does not extend beyond that hence I am not convinced the paper is ready for publication at NeurIPS

final rating: borderline accept
The authors have addressed many of my concerns ( see rebuttal and answers below )

---

> ### Author Rebuttal · Authors · 2024-08-07
>
> Thank you for the issues you raised!  We address your concerns below:
>
> ---
> **W1: Limited novelty and scope.**
>
> First, we humbly clarify that this work does deliver new technical insight, which is “**reducing fragment-level redundancy, in addition to commonly used primitive-level redundancy, enables a better trade-off between the rendering speed and quality**.” In addition, compared to existing 3DGS pruning works, which prune entire Gaussian primitives, we are the first to explore redundancy at the fragment level. Hence, our approach is complementary to existing works and serves as an effective plug-in module of easy use.
> As such, we believe our insight and technique could extend beyond 3D Gaussian Splatting to other works that utilize a rasterization-based rendering pipeline, such as point clouds [a] and surface splatting [b,c].
>
> Second, 3D Gaussian-based rendering techniques have not only achieved superior performance in scene reconstruction and rendering but have also been successfully applied to various other 3D vision-related applications, such as animatable human avatars [d], scene editing [e], open vocabulary querying of dynamic scenes [f], and even physical simulation [g]. Given their success in these downstream tasks, we believe that fully optimizing the efficiency of the 3D Gaussian-based rendering pipeline is a crucial and timely research question, which can largely facilitate the deployment of these 3D Gaussian-powered applications on edge devices like AR/VR headsets and smartphones.
>
> Given the promise of 3DGS, our technique can serve as a plug-and-play add-on, allowing faster rendering speed without sacrificing rendering quality, thus promising to benefit the entire 3D community. We demonstrate this by extending our work to 4D Gaussian Splatting [29] (see Tab. 3 of our paper) and 3DGS-MCMC [h] (attached at the end of our response).
>
>
> [a] TRIPS: Trilinear Point Splatting for Real-Time Radiance Field Rendering, Eurographics 2024
>
> [b] Surface Splatting, SIGGRAPH 2001
>
> [c] Differentiable surface splatting for point-based geometry processing, ToG 2019
>
> [d] D3GA - Drivable 3D Gaussian Avatars, Arxiv 2023
>
> [e] GaussianEditor: Swift and Controllable 3D Editing with Gaussian Splatting, CVPR 2024
>
> [f] LangSplat: 3D Language Gaussian Splatting, cvpr 2024
>
> [g] PhysGaussian: Physics-Integrated 3D Gaussians for Generative Dynamics, CVPR 2024
>
> [h] 3D Gaussian Splatting as Markov Chain Monte Carlo, Arxiv 2024
>
> ---
> **W2: Extra training cost not reflected in Tab. 1 and 2.**
>
> Thank you for the suggestion! For Tab. 1 in our paper, we would like to clarify that it presents the profiling experiments conducted on top of existing works, thus incurring no additional training cost. For Tab. 2, we have summarized the corresponding additional training cost (measured on an RTX A5000 GPU) of our method compared to the baselines, as shown in the table below.
>
> | Dataset   | On top of Unpruned 3D Gaussians [16] | On top of Pruned 3D Gaussians [19]       |
> |-|-|-|
> | Mip-NeRF 360 [6]| 7.68 min (17% of [16]’s training time)| 3.43 min (20% of [19]’s training time) |
> | Tanks & Temples [20]| 4.35 min (16% of [16]’s training time)| 2.17 min (19% of [19]’s training time) |
> | Deep Blending [23]| 7.11 min (19% of [16]’s training time)| 3.41 min (21% of [19]’s training time) |
>
>
> We can see that the additional training cost adds less than 25% overhead compared to the baselines, as we optimize opacity and truncation thresholds for less than 30% of the iterations in the second training stage. Furthermore, if the goal is to achieve rendering quality on par with the baselines, we can further reduce the number of iterations in the second stage of our method (e.g., reducing from 5,000 to 1,000 iterations), achieving 1.7× higher FPS while incurring only 3.5% additional training cost and maintaining the same rendering quality (i.e., PSNR/SSIM) on the Mip-NeRF 360 dataset compared to native 3DGS. We will include this discussion in the final version of our paper.
>
> ---
> **W3: Why increased PSNR after pruning? Naive optimization: easily reduce the opacity to 0 instead of pruning it.**
>
> Good question! In a nutshell, the PSNR gain achieved by our work stems from the ability of a Gaussian function with a cutoff to better fit sharp signals in real-world scenes compared to the native Gaussian function, as shown in Fig. 1 of the global 1-page response. Specifically, we would like to clarify the following two points:
>
> **P1: Enhancing the Gaussian function with learnable thresholds increases its ability to fit signals.**
> As illustrated in Fig. 1(a), enhancing each Gaussian function with a learnable threshold essentially gives each Gaussian a flexible cutoff range. In Fig. 1(c), we show that the truncated Gaussian function better fits various signals as compared to the native Gaussian functions, thanks to the enhanced capability of the learnable threshold/cutoff. Since sharp signals widely exist in real-world scenes and are the primary motivation behind existing anti-aliasing works [6,7], better fitting these signals using a Gaussian with a cutoff, as in our method, improves the PSNR by an average of 0.16 on the Tanks & Temples dataset, as shown in Tab. 2 of our paper.
>
> **P2: Mathematically, reducing the opacity of Gaussians is NOT equivalent to fragment pruning.**
> As illustrated in Fig. 1(a) and Fig. 1(b), we demonstrate that fragment pruning (i.e., a learnable cutoff of the Gaussian function) is not mathematically equivalent to lowering the opacity (i.e., a global magnitude factor) of a Gaussian function. The former impacts only the Gaussian value around the cutoff threshold, while the latter reduces the magnitude of the Gaussian function globally. Therefore, the Gaussian with cutoffs adopted in our method can better fit sharp signals, such as square signals, compared to “easily reduce the opacity to 0”.

---

> > ### Comment · Reviewer_7Aqo · 2024-08-09
> >
> > I thank the authors for the great rebuttal. I still have concerns about why are we increasing the FPS of 3DGS. it is already real-time on many available devices, Jetson Orin shown in the rebuttal is it a robotics device not a home gadget? can we do on CPUs is the actual question? I see memory and optimization time as a more challenging aspect of 3DGS, do the authors agree?
> >
> > I will increase my score to borderline accept.

---

> > > ### Author Response · Authors · 2024-08-11
> > >
> > > Dear Reviewer 7Aqo,
> > >
> > > Thank you for your thoughtful review and positive feedback on our rebuttal! We are encouraged to hear that our rebuttal has addressed most of your concerns and appreciate the opportunity to clarify the remaining points.
> > >
> > >
> > > **P1: Jetson Orin shown in the rebuttal is a robotics device, not a home gadget.**
> > >
> > > Thank you for raising this point! While it is true that the Jetson series GPUs, including the Orin, are often associated with robotics, they have also been extensively used in consumer-grade edge AR/VR devices, where real-time 3D rendering is highly desirable. For example, the Magic Leap 1 glasses [i] utilize the Jetson TX2 [j], and the Magic Leap 2 glasses [l] are powered by the IGX Orin [m], which offers a similar number of CUDA cores and tera floating-point operations per second (TFLOPs) as the Jetson AGX Orin [o]. As a result, Jetson-series GPUs have become a popular choice for AR/VR algorithm benchmarking [p] and deployment [q, r]. Therefore, our profiling and benchmarking results on the Jetson Orin NX can serve as a helpful baseline for future innovations and also provide valuable insights for the real-world deployment of 3DGS algorithms on edge AR/VR devices.
> > >
> > >
> > > [i] Magic Leap 1, Magic Leap 2019
> > >
> > > [j] Jetson TX2 Module, Nvidia 2017
> > >
> > > [l] NVIDIA IGX + Magic Leap 2 XR Bundle, Magic Leap 2024
> > >
> > > [m] IGX Orin, Nvidia 2023
> > >
> > > [o] Jetson AGX Orin, Nvidia 2023
> > >
> > > [p] ILLIXR: Enabling End-to-End Extended Reality Research, IISWC 2021
> > >
> > > [q] Deja View: Spatio-Temporal Compute Reuse for Energy-Efficient 360° VR Video Streaming, ISCA2020
> > >
> > > [r] Edge Assisted Real-time Object Detection for Mobile Augmented Reality, MobiCom 2019
> > >
> > >
> > > **P2: Can we do on CPUs is the actual question?**
> > >
> > > We agree that enabling real-time rendering on CPUs is an intriguing research direction. However, CPUs are primarily designed for fast sequential processing [s] and are not optimized for the massively parallel processing required for tasks like parallel pixel/fragment rendering. This is why modern AR/VR devices predominantly rely on GPUs for graphics rendering [i,l]. For instance, a 105-watt AMD Ryzen 5 7600X CPU [t] can only achieve a peak performance of approximately 0.09 TFLOPs, which is significantly lower than the 1.88 TFLOPs on the 15-watt Orin NX edge GPU. Consequently, rendering 3DGS in real-time on CPUs is currently not feasible due to the vast number of fragments involved and the associated computational demands. For example, to achieve real-time rendering (i.e., over 60 FPS) in the Kitchen scene of the Mip-NeRF 360 dataset [6], the rasterization stage alone would require around **0.11 TFLOPs**, which is beyond the **0.09 TFLOPs** provided by a Ryzen 5 7600X CPU.
> > >
> > >
> > > [s] Computer Architecture: A Quantitative Approach, Morgan Kaufmann 2017
> > >
> > > [t] AMD Ryzen 5 7600X, Advanced Micro Devices 2022
> > >
> > >
> > > **P3: Why are we increasing the FPS of 3DGS? It is already real-time on many available devices. I see memory and optimization time as a more challenging aspect of 3DGS, do the authors agree?**
> > >
> > > We agree that reducing memory usage and optimization time are important and challenging goals. However, further improving 3DGS’s rendering speed is also crucial, as AR/VR applications typically require real-time rendering speeds of 60+ FPS to ensure an immersive user experience [u, v]. While 3DGS achieves real-time performance on many server-grade (i.e., high-end) GPUs, such as the A100 and RTX 3090, our experiments indicate that even with state-of-the-art pruning methods like Mini-Splatting [19], the rendering speed remains limited (e.g., 35 FPS on the Mip-NeRF 360 dataset [6]) on resource-constrained edge computing platforms, such as the Jetson Orin NX. This underscores both the need and the challenge of further enhancing the rendering efficiency of 3DGS in order to enable their applications in numerous edge rendering applications, such as robotics and AR/VR.
> > >
> > >
> > > [u] Is it Real? Measuring the Effect of Resolution, Latency, Frame rate and Jitter on the Presence of Virtual Entities, ISS 2019
> > >
> > > [v] Realistic Rendering at 60 Frames Per Second — Past, Present, and Future, HPG 2024

---

> ### Author Response · Authors · 2024-08-07
> **Apply our method on top of GS-MCMC @ Mip-NeRF 360 [6]**
>
> |        | PSNR  | SSIM | LPIPS | FPS | Train Time (min)|
> |--------|-------|------|-------|-----|----|
> | GS-MCMC| 29.72 | 0.89 | 0.19  | 20  | 69.03 |
> | GS-MCMC + Ours  | 29.75 | 0.90 | 0.19  | 36  | 69.03 +10.11  |
>
> *Averaged on the Mip-NeRF 360 [6] dataset. FPS Measured on Orin NX; train time measured on A5000.*

---

### Official Review · Reviewer_TxBU · 2024-07-02

**Soundness:** 3
**Presentation:** 3
**Contribution:** 3
**Rating:** 6
**Confidence:** 4

**Summary:**

This paper introduces a novel approach to accelerate rendering speed in 3D Gaussian Splatting (3DGS) by selectively pruning overlapping fragments. This technique serves as an orthogonal enhancement to existing pruning methods (focuses on reducing the number of primitives) by selectively pruning overlapping fragments, thereby significantly accelerating rendering. The proposed adaptive pruning framework dynamically optimizes pruning thresholds for each Gaussian fragment, leading to marked improvements in both rendering speed and quality.

**Strengths:**

1. Through a detailed analysis of the efficiency bottleneck of the rendering pipeline and the impact of the number of Gaussian primitives on efficiency, this paper introduces a novel concept of fragment pruning to enhance rendering efficiency in 3D Gaussian Splatting (3DGS).

2. Dynamically optimizes pruning thresholds for each Gaussian fragment, leading to significant improvements in rendering speed and quality.

3. The experimental results in both static and dynamic scenes demonstrate the effectiveness of the proposed fragment pruning method in enhancing rendering efficiency without compromising visual fidelity.

**Weaknesses:**

1. The experimental section of the paper includes only Mini-Splatting [1] as a state-of-the-art Gaussian primitive pruning method among the baselines in static scenes' evaluations, while other comparisons are based on raw rendering pipelines. In dynamic scenes' evaluations, there no comparison with any state-of-the-art Gaussian primitive pruning techniques. While this comparison demonstrates the effectiveness of the proposed method, it lacks a demonstration of its advancement over other state-of-the-art Gaussian primitive pruning methods. Including more state-of-the-art Gaussian primitive pruning methods could enhance the persuasiveness of the evaluation.

2. The baseline selection of the static scenes is unreasonable. In the experimental section (Table 2), among the four rendering pipelines, only one is rasterization-based radiance field rendering method. Since the focus of this paper is on pruning based on 3D Gaussian Splatting (3DGS) representation, the comparison should primarily involve rasterization-based radiance field rendering methods similar to 3DGS.

[1] Guangchi Fang and Bing Wang. Mini-splatting: Representing scenes with a constrained number of gaussians. arXiv preprint arXiv:2403.14166, 2024.

**Questions:**

1. Why did you primarily compare with neural radiance field methods instead of rasterization-based radiance field rendering methods as your baseline comparison in the rendering pipeline?

2. Why did you include only Mini-Splatting [1] without comparing with more state-of-the-art Gaussian primitive pruning techniques? Why was there no comparison with any state-of-the-art Gaussian primitive pruning techniques in dynamic scenes?

[1] Guangchi Fang and Bing Wang. Mini-splatting: Representing scenes with a constrained number of gaussians. arXiv preprint arXiv:2403.14166, 2024.

**Limitations:**

The authors have already mentioned the limitations regarding training time. Additionally, this method may also have limitations when pruning other types of primitive representation.

---

> ### Author Rebuttal · Authors · 2024-08-07
>
> Thank you for raising questions regarding our baseline choices! We provide the following clarifications:
>
> ---
> **W1 & Q2: No comparison with other state-of-the-art Gaussian pruning pipelines on static scenes.**
>
> Following your suggestion, we have added the comparison with additional Gaussian primitive pruning methods on the Mip-NeRF-360 dataset in the table below.
>
> | Method                      | PSNR | SSIM  | LPIPS | FPS (On Orin NX) |
> |-----------------------------|------|-------|-------|------------------|
> | LightGaussian [35]           | 27.28| 0.805 | 0.243 | 28               |
> | Compact 3D Gaussian [32]     | 27.08| 0.798 | 0.247 | 24               |
> | LP-3DGS [a] (RadSplat Score)| **27.47**| 0.812 | 0.227 | 24               |
> | Mini-Splatting [19]          | 27.34| **0.822** | 0.217 | 35          |
> | Mini-Splatting [19] + Ours       | 27.38| **0.822** | **0.209** | **54**      |
>
>
>
> We observe that (1) among all prior Gaussian primitive pruning methods, Mini-Splatting [19] achieved the best trade-off between rendering quality and speed, and (2) our method, which performs pruning at the fragment level, further improves both the rendering quality and speed of Mini-Splatting [19]: +0.04 PSNR and 1.54× faster rendering speeds.
>
> [a] LP-3DGS: Learning to Prune 3D Gaussian Splatting, Arxiv 2024
>
>
> ---
> **W2 & Q2: Lack of comparison with other Gaussian primitive pruning works on dynamic scenes.**
>
> We would like to clarify that, to the best of our knowledge, existing Gaussian primitive pruning works focus exclusively on *static* 3D Gaussians. We are not aware of any work that performs pruning on dynamic 3D Gaussians. Our work is the first post-processing pruning framework designed to improve rendering speed on dynamic scenes. Consequently, our experiments on dynamic scenes were conducted on top of RT-4DGS [29], the state-of-the-art open-source 3DGS for dynamic scenes in terms of rendering quality and speed trade-off. As shown in Table 3 of our paper, our method achieved 1.57× faster rendering speeds with a 0.03 PSNR improvement compared to RT-4DGS [29].
>
>
>
> ---
> **Q1: Why did you primarily compare with neural radiance field methods instead of rasterization-based radiance field rendering as your baseline?**
>
>
> We did not primarily compare our work with rasterization-based methods due to their significantly lower rendering quality (e.g., more than 3 PSNR lower, as shown in the table below) compared to NeRF-based methods and native 3DGS. This baseline selection strategy aligns with that used in native 3DGS. Meanwhile, following your suggestion, we have included a comparison with additional rasterization-based methods in the table below.
>
> | Method                        | PSNR  | SSIM  | LPIPS | FPS (On Orin NX)           |
> |-------------------------------|-------|-------|-------|----------------------------|
> | MobileNeRF [14]               | 21.95 | 0.470 | 0.470 | 60                         |
> | NeRF2Mesh [b] (render w/ mesh) | 22.36 | 0.493 | 0.478 | 60                         |
> | NeRFMeshing [c]               | 23.59 | -     | -     | Code not released          |
> | Mini-Splatting [19]           | 27.34 | **0.822** | 0.217 | 35                         |
> | Mini-Splatting [19] + Ours    | 27.38 | **0.822** | **0.209** | 54                   |
>
>
> We can see that despite the faster rendering speeds (> 60 FPS), the rasterization-based method fails to achieve the same rendering quality as the native 3DGS. In contrast, our method not only achieves approximately 60 FPS rendering speeds but also slightly improves the rendering quality (+0.04 PSNR).
>
> [b] Delicate Textured Mesh Recovery from NeRF via Adaptive Surface Refinement, ICCV 2023
>
> [c] Distilling Neural Radiance Fields into Geometrically-Accurate 3D Meshes, 3DV 2024

---

> > ### Comment · Reviewer_TxBU · 2024-08-07
> > **The response to Weakness 1 and Question 2 has addressed some of my concerns. However, the authors seem to have misunderstood Weakness 2 and Question 1, and their response is somewhat confusing.**
> >
> > Thank you for providing additional comparisons with other Gaussian primitive pruning methods, which to some extent addresses my second question.
> >
> > However, it seems that the authors have misunderstood my question 1. NeRF represents neural radiance field methods, while 3DGS is inherently a rasterization-based radiance field method. The authors' response to my comment on weakness 2 and issue 1 is confusing. Therefore, I would like to restate my question here: The baseline selection of the static scenes is unreasonable. In the experimental section (Table 2), among the four rendering pipelines, only one is rasterization-based radiance field rendering method. Since the focus of this paper is on pruning based on 3D Gaussian Splatting (3DGS) representation, the comparison should primarily involve rasterization-based radiance field rendering methods (3DGS-based). Why did you primarily compare with neural radiance field methods (NeRF-based) instead of rasterization-based radiance field rendering methods (3DGS-based) as your baseline comparison in the rendering pipeline?
> >
> > In the response, it seems the authors misunderstood rasterization-based rendering pipelines as mesh-based rendering pipelines, thus the provided supplementary comparisons did not address my question. I encourage the authors to provide a quantitative evaluation of the proposed pruning method applied to various SOTA 3DGS pipelines. Additionally, a quantitative comparison of the proposed method with other SOTA Gaussian pruning methods across multiple SOTA 3DGS pipelines should be included. Setting almost all baselines as NeRF-based methods for comparison is insufficient because 3DGS itself significantly surpasses NeRF in rendering speed.

---

> ### Author Response · Authors · 2024-08-08
>
> Dear Reviewer TxBU,
>
> Thank you for reviewing our rebuttal and for your prompt response! We are pleased to hear that our response has addressed some of your concerns.
>
> We greatly appreciate your further clarification on Question 1 and your constructive suggestions. It is worth noting that during the rebuttal period, we successfully extended our framework to 3DGS-MCMC [d], a 3DGS variant with better rendering quality than the vanilla 3DGS [16]. The corresponding results on the Mip-NeRF 360 dataset [6] have been included in the last table of our response to Reviewer ytrw and are listed below.
>
> |        | PSNR  | SSIM | LPIPS | FPS (On Orin NX) |
> |--------|-------|------|-------|-----|
> | GS-MCMC| 29.72 | 0.89 | 0.19  | 20  |
> | GS-MCMC + Ours  | 29.75 | 0.90 | 0.19  | 36  |
>
> We will continue to extend our pruning framework to additional 3DGS pipelines and provide quantitative evaluations as you suggested. **We will keep you updated on our progress during this discussion period.**
>
> Thank you again for your valuable feedback.
>
>
> [d] 3D Gaussian Splatting as Markov Chain Monte Carlo, ArXiv 2024

---

> ### Author Response · Authors · 2024-08-13
> **Additional Benchmark Results across More 3DGS-based Pipelines #1**
>
> Thank you once again for taking the time to read our rebuttal and further providing your thoughtful feedback! In response to your suggestion, we have extended our evaluation to include three rasterization-based (3DGS-based) rendering pipelines that enhance 3DGS rendering quality from different aspects:
>
> + **Mip-Splatting [e]**: A 3DGS-based pipeline for **anti-aliasing** in extreme camera locations (e.g., very close to objects).
>
> + **GS-MCMC [d]**: A 3DGS-based pipeline with a more principled approach to densifying Gaussian primitives, resulting in **fewer artifacts (e.g., floaters)** in the reconstructed scenes.
>
> + **RAIN-GS [f]**: A 3DGS-based pipeline that **relies less on accurate point cloud initialization**, utilizing a dedicated point cloud optimization process.
>
> ***
>
> **Benchmark Results on Mip-Splatting [e] on the Mip-NeRF 360 dataset [6]:**
> |                                 | PSNR  | SSIM  | LPIPS | FPS (Orin NX) |
> |---------------------------------|-------|-------|-------|------------------|
> | Original* [e]                   | 27.88 | ***0.837*** | 0.175 | 21               |
> | + LightGaussian [35]            | 27.89 | 0.834 | 0.188 | 23               |
> | + LP-Gaussian [a] (Radsplat Score [i]) | 27.87 | 0.834 | 0.189 | 22        |
> | + Ours                           | ***27.93*** | ***0.837*** | ***0.172*** | 30               |
> | + LightGaussian [35] + Ours     | ***27.93*** | 0.835 | 0.185 | ***33***              |
>
> *\*: Note that the official Mip-Splatting codebase has been enhanced with an improved densification process [j], resulting in higher rendering quality than what was originally reported in its paper. We used the single-scale training and single-scale testing setting of Mip-Splatting [e].*
>
> ***
> **Benchmark Results on GS-MCMC [d] on the Mip-NeRF 360 dataset [6]:**
> |                                      | PSNR  | SSIM  | LPIPS | FPS (Orin NX) |
> |--------------------------------------|-------|-------|-------|------------------|
> | Original* [d]                        | 29.72 | 0.886 | 0.188 | 20               |
> | + LightGaussian [35]                 | 29.59 | 0.888 | 0.201 | 22               |
> | + LP-Gaussian [a] (Radsplat Score [i]) | 29.67 | 0.897 | 0.181 | 23             |
> | + Ours                               | 29.75 | 0.895 | 0.186 | 36               |
> | + LP-Gaussian [a] + Ours             | **29.86** | **0.899** | **0.176** | **45**  |
>
> *\*: We used the random point cloud initialization setting of GS-MCMC [d].*
>
> ***
> **Benchmark Results on RAIN-GS [f] on the Mip-NeRF 360 dataset [6]:**
> |                                      | PSNR  | SSIM  | LPIPS | FPS (Orin NX) |
> |--------------------------------------|-------|-------|-------|------------------|
> | Original* [f]                        | 27.23 | **0.807** | 0.229 | 18           |
> | + LightGaussian [35]                 | 27.33 | 0.805 | 0.238 | 24               |
> | + LP-Gaussian [a] (Radsplat Score [i])                    | 27.28 | 0.805 | 0.231 | 27               |
> | + Ours                               | 27.26 | 0.804 | **0.227** | 34        |
> | + LightGaussian [35] + Ours          | **27.36** | 0.806 | 0.231 | **40**       |
>
> *\*: We used the random point cloud initialization setting of RAIN-GS [f].*
>
> With the addition of the three extra rendering pipelines, we benchmarked our fragment pruning method against two state-of-the-art plug-and-play pruning methods: **LightGaussian [35]** and **LP-3DGS [a]**, as shown in rows 3-4 in the above tables. In addition to directly comparing our method with these baseline pruning methods, we also applied our method on top of the baseline pruning method which provides better rendering quality (see row 6 of the above tables) to demonstrate its compatibility with existing primitive-based pruning techniques. We would like to note that Mini-Splatting [1] was not included in this set of additional experiments due to insufficient time during the discussion period to integrate its end-to-end, fully customized optimization pipeline (i.e., densification, reinitialization, and simplification techniques) with the customized optimization pipelines of those 3DGS variants [e, d, f]. We will continue this experiment and include it in our final version.

---

> ### Author Response · Authors · 2024-08-13
> **Additional Benchmark Results across More 3DGS-based Pipelines #2**
>
> Based on the benchmark results summarized in the tables above, we can draw the following conclusions:
>
> + **Original Rendering Pipelines [e,d,f] vs. Ours**: Our proposed fragment pruning method consistently enhances rendering speed without compromising rendering quality across various 3DGS-based rendering pipelines. Specifically, the proposed method (Original + Ours) increases rendering speed by 1.4x, 1.8x, and 1.9x on Mip-Splatting [e], GS-MCMC [d], and RAIN-GS [f], respectively, with a 0.03 to 0.05 higher PSNR.
>
> + **Baseline Pruning Techniques [35, a] vs. Ours**: Our proposed fragment pruning not only achieves better accuracy vs. efficiency trade-offs than baseline primitive-level pruning techniques [35, a], e.g., 0.08 to 0.16 higher PSNR and 1.5x to 1.6x faster rendering speeds on GS-MCMC [d], but also can further improve these existing primitive-level pruning techniques when applied on top of them,e.g., 0.19 to 0.27 higher PSNR and 1.9x to 2.0x faster rendering speeds on GS-MCMC [d].
>
> The conclusions above are consistent with our observations on the vanilla 3DGS as discussed in our submitted manuscript: our proposed method (1) provides **better accuracy vs. efficiency trade-offs** than baseline rendering pipelines and (2) is **complementary** to existing primitive-based pruning techniques. We greatly appreciate your constructive suggestions and believe adding these additional baselines and experiments will help strengthen our work and contributions.
>
> [e] Mip-Splatting: Alias-free 3D Gaussian Splatting, CVPR 2024
>
> [f] Relaxing Accurate Initialization Constraint for 3D Gaussian Splatting, Arxiv 2024
>
> [i] RadSplat: Radiance Field-Informed Gaussian Splatting for Robust Real-Time Rendering with 900+ FPS, Arxiv 2024
>
> [j] Gaussian Opacity Fields: Efficient High-quality Compact Surface Reconstruction in Unbounded Scenes, Arxiv 2024

---

### Official Review · Reviewer_RCBw · 2024-07-12

**Soundness:** 4
**Presentation:** 3
**Contribution:** 3
**Rating:** 6
**Confidence:** 5

**Summary:**

This paper investeigates the relationship between primitive pruning and rendering speed. It identifies that pruning 3D primitives does not translate proportionally to higher rendering speed and shows the real measure affecting rendering speed is the number of fragments (projected splats partaking in pixel color rendering). Finally it proposes a simple idea for optimizing Gaussian opacity truncation threshold, rather than setting a fixed value that reduces the number of fragments and results in higher rendering speed while keeping compression factor and quality.

**Strengths:**

- I like the idea behind this paper as it does a thorough analysis on the effect of pruning methods for compression on the rendering speed. It identifies why compression is not proportionally resulting in higher rendering speed and then addresses the issue with a simple and effective idea.
- The manually set threshold parameter in 3DGS is one of the many hyper-parameters of this method that need fine-tuning and probably is not optimal. Making this value trainable is an important contribution.
- The method shows significant improvement in rendering speed while maintaining or slightly improving both the compression factor and the quality.

**Weaknesses:**

- I would love to see experiment showing rendering speed and PSNR for different fixed values of threshold vs the learnt threshold. This is to verify setting the threshold to a higher value manually (and training with that) has less improvement than the proposed method. This would effectively show that having per Gaussian optimized threshold is truly needed.
- I think having a longer training time for this purpose is valid. However, I am curious as to why is the method posed as a post processing? Did you explore doing an alternating optimization between the usual 3DGs parameters and the thresholds during training? I am not sure if that would result in lower training time in total than the usual training + post-processing, but it is interesting to see.
- Additional point cloud renderings to assess the density of points/fragments on the qualitative figures of the supplementary material is helpful to have.
- The results for overfitting with training on all the 3DGS parameters is not shown in the supplemental material. It is better to have a visualization for that experiment in the supplementary for reference.

**Questions:**

please see above (weaknesses) for some questions.

**Limitations:**

The limitation is adequetly addressed.

---

> ### Author Rebuttal · Authors · 2024-08-07
>
> We appreciate your recognition of our work and your suggestions to further enhance our experimental results.
>
> ---
> **W1: I would love to see an experiment showing rendering speed and PSNR for different fixed values of threshold vs the learnt threshold. This would effectively show that having per Gaussian optimized threshold is truly needed.**
>
> We conducted a grid search on fixed threshold values and followed the same fine-tuning protocol as described in Section 6.2 of our paper. The results summarized in Fig. 3 (b) of the global 1-page response indicate that our method achieves a better accuracy vs. efficiency trade-off than tuning fixed threshold values, highlighting that learning a per-Gaussian threshold, as in our method, is truly necessary.
>
>
> ---
> **W2: Did you explore doing an alternating optimization between the usual 3DGs parameters and the thresholds during training? I am not sure if that would result in lower training time in total than the usual training + post-processing, but it is interesting to see.**
>
> Thank you for this  insightful question! Following your suggestion, we conducted an alternating optimization experiment on the MipNeRF 360 dataset [6]. Specifically, we maintained the training process for the first 15,000 iterations as in the vanilla 3D Gaussian Splatting [16], then alternated between threshold learning and 3D Gaussian parameter learning every 5,000 iterations up to 40,000 iterations. We tested the PSNR every 5,000 iterations, and the results are shown in the table below:
>
>
> | Iteration  | PSNR (Bicycle) | PSNR (Garden) | PSNR (Kitchen) | PSNR (Room) | Avg   |
> |------------|----------------|---------------|----------------|-------------|-------|
> | 15000      | 24.47          | 26.43         | 30.27          | 30.23       | 27.85 |
> | 20000      | 25.06          | 27.07         | 30.96          | 30.83       | 28.48 |
> | 25000      | 25.07          | 27.05         | 30.99          | 30.86       | 28.49 |
> | 30000      | 25.07          | 27.06         | 31.00          | 30.87       | 28.50 |
> | 35000      | 25.07          | 27.05         | 31.00          | 30.88       | 28.50 |
> | 40000      | 25.07          | 27.05         | 31.00          | 30.88       | 28.50 |
> | 3DGS (30000)| 25.19         | 27.30         | 31.29          | 31.45       | 28.81 |
> | Ours (35000)| 25.21         | 27.33         | 31.49          | 31.68       | 28.93 |
>
>
> We can observe that the PSNR quickly saturates at around 20,000 iterations and fails to match the PSNR of the vanilla 3D Gaussian and our method. We conjecture that the lower rendering quality in this setting, compared to our method, is due to reducing the covered pixels per Gaussian too early in the training stage, which limits the number of pixels from which it can receive gradients. In contrast, our method strikes a balance between ease of use and achieved performance compared to alternating optimization. However, we believe that the alternating optimization you proposed has the potential to achieve superior rendering quality with more thorough parameter tuning. We will continue to explore this approach and include a discussion on optimization methods in our final version.
>
> ---
> **W3: Additional point cloud renderings to assess the density of points/fragments on the qualitative figures of the supplementary material are helpful to have**
>
> We have added the per-pixel fragment density in Fig. 2 of the global 1-page response, demonstrating the effectiveness of our proposed method in reducing fragment density across different pixels (e.g., reducing the average fragments per pixel from 49.2 to 32.9 in the Bicycle scene [6]), thereby improving overall rendering speed. We will include this discussion and provide additional visualizations of the point cloud renderings in our final version.
>
>
> ---
> **W4: The results for overfitting with training on all the 3DGS parameters are not shown in the supplemental material.**
>
> Thank you for the suggestion! We conducted overfitting experiments by loading pre-trained checkpoints at 30,000 iterations from native 3DGS and continuing training for an additional 20,000 iterations following the native 3DGS settings. As shown in Fig. 3 (c) of the global 1-page response, while overfitting does improve the overall PSNR, it still results in a slightly lower PSNR (e.g., -0.03) compared to our method, even with 40% more iterations.

---

> ### Comment · Area_Chair_yjL4 · 2024-08-10
> **Please consider author rebuttal & enter discussion with authors!**
>
> Dear Reviewer,
>
> I wanted to gently remind you to please review the rebuttal provided by the authors. Please enter a discussion with the authors to address any outstanding questions.
>
> Your feedback is invaluable to the decision-making process, and if you feel that the rebuttal addresses any of your concerns, please consider updating your score accordingly.
>
> Thank you for your continued dedication to ensuring a fair and thorough review process!
>
> Best, Your AC

---

> ### Comment · Reviewer_RCBw · 2024-08-11
>
> I would like to thank the authors for their response to my concerns, I have no other major concerns. I will retain my original score, as I believe the paper is technically sound, gives a valuable analysis and will have a moderate-to-high impact on the 3D rendering area.

---

> > ### Author Response · Authors · 2024-08-11
> >
> > Thank you very much for taking the time to review our rebuttal and for your positive feedback! We are encouraged by your recognition of the technical soundness and potential impact of our work.

---

### Official Review · Reviewer_ytrw · 2024-07-12

**Soundness:** 3
**Presentation:** 3
**Contribution:** 3
**Rating:** 6
**Confidence:** 5

**Summary:**

The paper presents a pruning method to improve the time efficiency of 3D Gaussian Splatting. Different from existing methods that prune Gaussians, the proposed method prunes pixels per each Gaussian. Specifically, for each pixel ray, the proposed method reduces the number of Gaussians needed to compose the color. To this end, the paper proposes to learn the Gaussian-wise threshold to adaptively prune the unnecessary Gaussians for each pixel ray. Different from the fixed global threshold in existing 3DGS pipelines, their per-gaussian threshold effectively decreases the number of Gaussians without degrading the rendering quality. The paper validates their method in different datasets and two devices, showing the advantages over previous methods that try to improve the time efficiency.

**Strengths:**

The work has a certain impact on the applications of 3DGS. Because it’s simple and effective for improving the time efficiency of 3DGS.  To be more specific, the proposed method acts as an additional step that improves the inference efficiency of all trained 3D Gaussian Splats with degraded rendering quality.  This would be very useful for industrial applications.

The profiling results are useful to the community. This helps in understanding the latency distribution over steps of 3DGS.

The paper is also easy to follow.

**Weaknesses:**

The paper has several weaknesses which I’ll detail below

Writing:
- The title does not match what the paper achieves. In my understanding, the proposed method does not change the number of Gaussians. So it does not make sense to claim that the Gaussians can be sparser.
The concept of fragment is not easy to follow in the abstract and intro. From my personal viewpoint, the author could simply say the Gaussians per pixel ray.
- In the abstract, it would be better to always mention the device/platform when the paper claims the improved speedup. At line 35 of the intro section, similarly, it would be better to be more specific.
- The implementation details of fine-tuning have the loss description missing. Is that l1 loss and SSIM loss?
- Also, what’s the connection between the proposed method and two staged frameworks to accelerate NeRF -- e.g., NeRF to mesh?  Intuitively, they share the same idea of reducing the number of samples per ray. In which case, I believe it makes sense to discuss the connection between these two methods.

Methods:
- I don’t quite understand the insight of fine-tuning to learn the Gaussian-wise threshold. Despite the improved results, I’m not sure why the method can automatically discover the perfect Gaussian-wise threshold that reduces the number of Gaussians per pixel ray. Theoretically, it’s also possible to learn the thresholds that increase the number of Gaussians per pixel ray. It would be better to elucidate the method with more insights.

Experiments:
- There are naive methods to encourage a smaller number of Gaussians per pixel. For example, we could encourage smaller scales of Gaussians.  It might also have the same effect as the proposed method.  Given the native nature of this simple idea, I would encourage the authors to simply discuss or compare it with this naive idea.
- An edge-device demo could make this paper stronger, thus I strongly recommend it. The reason is that Gaussian Splatting is real-time in GPU already. While it’s necessary to further improve it as it’s useful in some other applications, it might be more interesting to the research community if the proposed method can enable real-time efficiency in edge devices.
- From my point of view, there might be randomness in terms of profiling. So I disagree with the authors that the statistics are stable and error bars are not needed.
- Instead of Vanilla 3DGS in 3D and 4D settings, I strongly recommend authors validate their methods in other variants of 3DGS including Mip3DGS, GS-MCMC, etc. I would expect the reduced benefits of using the proposed methods without this validation.

**Questions:**

Please address the questions and concerns that I listed in the weaknesses section.

I currently tend to vote for borderline rejection as I think the paper has something that needs to be fixed. But I'm personally very positive about this paper. So I encourage authors to address the concerns that I have. Especially, I would like to see authors show the proposed solution to reduce the number of Gaussian per pixel ray is fully explored -- e.g., insight into why it can work, comparison with the naive solution, and so on.

-------

Final evaluation:
The rebuttal addressed most of the concerns. So I'm very happy to raise my score.

**Limitations:**

Yes, limitations and potential negative impacts have been discussed in the paper.

---

> ### Author Rebuttal · Authors · 2024-08-07
>
> We greatly appreciate your positive feedback and constructive suggestions for our work. Below, we address each of your comments in detail:
>
> ---
> **W1: Title mismatch: The method changes the # of Gaussians per pixel ray.**
>
> Thank you for the suggestion! We will revise the title accordingly and clarify that the projection of the 3D Gaussian onto the 2D image to be rendered is sparser, not the 3D Gaussian itself.
>
> ---
> **W2 & W3: Device details, # of Gaussians, and losses.**
>
> Thanks for pointing these out! To address your suggestions, the following details will be added to our final version:
> > + Line 18: The speedup is achieved on an edge GPU device Jetson Orin NX [18].
> > + Line 35: The concrete number of Gaussians is 3,161,131 on average in the Mip-NeRF 360 dataset [6] when trained using the original 3D Gaussian implementation [16].
> > + Sec. 6.2: We used the same loss function as the original 3D Gaussian Splatting paper [16] for static scenes (i.e., both SSIM Loss and L1 Loss) and the same loss function as the RT-4DGS paper [29] for dynamic scenes.
>
>
> ---
> **W4: Connection with two staged frameworks.**
>
> The connection between the prior two-stage works and our work can be summarized as follows:
>
> **Similarity**: Both prior works [14,a,b] and our work reduce computational workload by decreasing the computation per pixel ray.
>
> **Differences**: Our technique serves as an add-on to one of the most advanced scene representations, 3DGS, which allows for faster rendering without sacrificing quality. Different from prior two-stage works, our method does not alter the scene representation and thus can better maintain the rendering quality. As illustrated in Fig.2 of the global 1-page response, our method reduced the average number of fragments/Gaussians per pixel without strictly enforcing it to be 1. As a result, our approach can boost rendering speed while maintaining the rendering quality, achieving a +0.04 PSNR improvement on [6] over the vanilla 3DGS [16]. In contrast, the prior two-stage frameworks [14,a,b] convert NeRFs into meshes to enforce a single sample per pixel ray. This significant reduction, along with the differences between the scene representation used in training and rendering, leads to noticeable quality loss. For instance, on the MipNeRF-360 dataset [6], the works [14,a,b] achieved PSNR scores of 21.95/22.74/23.59, which are lower than even the vanilla NeRF [4] (23.85 PSNR, as reported by [6]).
>
> We will incorporate the discussion above into our final version.
>
>
> [a] Delicate Textured Mesh Recovery from NeRF via Adaptive Surface Refinement, ICCV 2023
>
> [b] Distilling Neural Radiance Fields into Geometrically-Accurate 3D Meshes, 3DV 2024
>
>
>
> ---
> **W5: Why can it automatically discover the perfect Gaussian-wise threshold to reduce the number of Gaussians per pixel ray? It could theoretically learn thresholds that increase the number as well.**
>
> Your understanding is correct! Learning the threshold can either decrease or increase the number of Gaussians per pixel ray. **The key insight behind this is that the decrease or increase is determined by the types of signals we want to represent using the Gaussian function**. Specifically, as shown in Fig. 1(a) of our global 1-page response, enhancing each Gaussian function with a learnable threshold essentially gives each Gaussian a flexible cutoff range. Thus, when the signal we want to fit has sharp edges (e.g., the square signal in Fig. 1(d) of our global 1-page response), learning the Gaussian-specific threshold offers the highest fidelity in fitting the ground truth signal compared to the original Gaussian function. Since the real-world scenes captured in our experiments contain many such sharp edges, also the source of anti-aliasing artifacts that [6,7] target, the threshold is learned to make the Gaussian cutoff narrower. This results in a reduced number of Gaussians per pixel ray. Our visualization of the number of fragments per pixel, with and without our method, in Fig. 2 of the global 1-page response also indicates that the number of fragments decreases most significantly at the edges of scenes.
>
> ---
> **W6: Naive methods: smaller scales of Gaussians.**
>
> Following your suggestions, we conducted experiments to encourage smaller scales of Gaussians by introducing an additional loss on scales. As summarized in Fig.3(a) of the global 1-page response, we found that while this approach can also improve rendering speed, it does not recover rendering quality to the same extent as our method. Specifically, the lower quality of the native method is due to the fact that smaller scales of Gaussians do not have the same effect as Gaussians with a cutoff, as illustrated in Fig. 1(a) and (c). This results in a failure to fit sharp signals, such as the square signal shown in Fig. 1(d) of the global 1-page response, as effectively as our method.
>
>
> ---
> **W7: Edge-device demo.**
>
> We have demonstrated the real-time edge-device demo setup on a 15-watt edge GPU device, the Nvidia Jetson Orin NX 16GB, as shown in Fig. 4(a) of the global 1-page response. We will include this setup as part of our code release in the final version.
>
> ---
> **W8: Profiling with  error bars.**
>
> Great suggestion! We have further enhanced our profiling in Fig. 4(b) of the global 1-page response by including the standard error.
>
>
> ---
> **W9: Validate our methods in other variants of 3DGS.**
>
> Thank you for the suggestion! Given the limited rebuttal time, we have extended our work to GS-MCMC [c]. As shown in the table below, our proposed method improves the rendering speed of GS-MCMC by 1.8× without compromising the rendering quality. We will conduct experiments on other 3D Gaussian variants, such as Mip3DGS, in the final version.
>
> |        | PSNR  | SSIM | LPIPS | FPS (On Orin NX)|
> |--------|-------|------|-------|-----|
> | GS-MCMC| 29.72 | 0.89 | 0.19  | 20  |
> | GS-MCMC + Ours  | 29.75 | 0.90 | 0.19  | 36  |
>
> [c] 3D Gaussian Splatting as Markov Chain Monte Carlo, Arxiv 2024

---

> > ### Comment · Reviewer_ytrw · 2024-08-10
> >
> > Thanks a lot to authors for their efforts in clarifying the questions/confusions! I think the rebuttal addressed most of questions that I have and I will adjust my evaluation accordingly.

---

> > > ### Author Response · Authors · 2024-08-11
> > >
> > > Thank you for taking the time to review our rebuttal responses and for providing positive feedback! We are encouraged to hear that our rebuttal has addressed most of your concerns.

---

> ### Author Response · Authors · 2024-08-13
> **Additional Benchmark Results on Other 3DGS Variants #1**
>
> Thank you once again for your constructive review and for providing your positive feedback on our initial rebuttal! We are writing to share our updated experimental results on your suggested **additional 3DGS variants**, beyond GS-MCMC [c] that was included in our previous response.
>
> Specifically, we have extended our evaluation to include three 3DGS variants that enhance 3DGS from different aspects:
>
> + **Mip-Splatting [e]**: A 3DGS variant for **anti-aliasing** in extreme camera locations (e.g., very close to objects).
> + **GS-MCMC [c]**: A 3DGS variant with a more principled approach to densifying Gaussian primitives, resulting in **fewer artifacts (e.g., floaters)** in the reconstructed scenes.
> + **RAIN-GS [f]**: A 3DGS variant that **relies less on accurate point cloud initialization**, utilizing a dedicated point cloud optimization process.
>
> ***
>
> **Benchmark Results on Mip-Splatting [e] on the Mip-NeRF 360 dataset [6]:**
> |                                 | PSNR  | SSIM  | LPIPS | FPS (Orin NX) |
> |---------------------------------|-------|-------|-------|------------------|
> | Original* [e]                   | 27.88 | ***0.837*** | 0.175 | 21               |
> | + LightGaussian [35]            | 27.89 | 0.834 | 0.188 | 23               |
> | + LP-Gaussian [d] (Radsplat Score [i]) | 27.87 | 0.834 | 0.189 | 22        |
> | + Ours                           | ***27.93*** | ***0.837*** | ***0.172*** | 30               |
> | + LightGaussian [35] + Ours     | ***27.93*** | 0.835 | 0.185 | ***33***              |
>
> *\*: Note that the official Mip-Splatting codebase has been enhanced with an improved densification process [j], resulting in higher rendering quality than what was originally reported in its paper. We used the single-scale training and single-scale testing setting of Mip-Splatting [e].*
>
> ***
> **Benchmark Results on GS-MCMC [c] on the Mip-NeRF 360 dataset [6]:**
> |                                      | PSNR  | SSIM  | LPIPS | FPS (Orin NX) |
> |--------------------------------------|-------|-------|-------|------------------|
> | Original* [c]                        | 29.72 | 0.886 | 0.188 | 20               |
> | + LightGaussian [35]                 | 29.59 | 0.888 | 0.201 | 22               |
> | + LP-Gaussian [d] (Radsplat Score [i]) | 29.67 | 0.897 | 0.181 | 23             |
> | + Ours                               | 29.75 | 0.895 | 0.186 | 36               |
> | + LP-Gaussian [d] + Ours             | **29.86** | **0.899** | **0.176** | **45**  |
>
> *\*: We used the random point cloud initialization setting of GS-MCMC [c].*
>
> ***
> **Benchmark Results on RAIN-GS [f] on the Mip-NeRF 360 dataset [6]:**
> |                                      | PSNR  | SSIM  | LPIPS | FPS (Orin NX) |
> |--------------------------------------|-------|-------|-------|------------------|
> | Original* [f]                        | 27.23 | **0.807** | 0.229 | 18           |
> | + LightGaussian [35]                 | 27.33 | 0.805 | 0.238 | 24               |
> | + LP-Gaussian [d] (Radsplat Score [i])                    | 27.28 | 0.805 | 0.231 | 27               |
> | + Ours                               | 27.26 | 0.804 | **0.227** | 34        |
> | + LightGaussian [35] + Ours          | **27.36** | 0.806 | 0.231 | **40**       |
>
> *\*: We used the random point cloud initialization setting of RAIN-GS [f].*
>
> With the addition of the three extra rendering pipelines, we benchmarked our fragment pruning method against two state-of-the-art plug-and-play pruning methods: **LightGaussian [35]** and **LP-3DGS [d]**, as shown in rows 3-4 in the above tables. In addition to directly comparing our method with these baseline pruning methods, we also applied our method on top of the baseline pruning method which provides better rendering quality (see row 6 of the above tables) to demonstrate its compatibility with existing primitive-based pruning techniques. We would like to note that Mini-Splatting [1] was not included in this set of additional experiments due to insufficient time during the discussion period to integrate its end-to-end, fully customized optimization pipeline (i.e., densification, reinitialization, and simplification techniques) with the customized optimization pipelines of those 3DGS variants [e, c, f]. We will continue this experiment and include it in our final version.

---

> ### Author Response · Authors · 2024-08-13
> **Additional Benchmark Results on Other 3DGS Variants #2**
>
> Based on the benchmark results summarized in the tables above, we can draw the following conclusions:
>
> + **Original Rendering Pipelines [e,c,f] vs. Ours**: Our proposed fragment pruning method consistently enhances rendering speed without compromising rendering quality across various 3DGS-based rendering pipelines. Specifically, the proposed method (Original + Ours) increases rendering speed by 1.4x, 1.8x, and 1.9x on Mip-Splatting [e], GS-MCMC [c], and RAIN-GS [f], respectively, with a 0.03 to 0.05 higher PSNR.
>
> + **Baseline Pruning Techniques [35, d] vs. Ours**: Our proposed fragment pruning not only achieves better accuracy vs. efficiency trade-offs than baseline primitive-level pruning techniques [35, d], e.g., 0.08 to 0.16 higher PSNR and 1.5x to 1.6x faster rendering speeds on GS-MCMC [c], but also can further improve these existing primitive-level pruning techniques when applied on top of them, e.g., 0.19 to 0.27 higher PSNR and 1.9x to 2.0x faster rendering speeds on GS-MCMC [c].
>
> The conclusions above are consistent with our observations on the vanilla 3DGS as discussed in our submitted manuscript: our proposed method (1) provides **better accuracy vs. efficiency trade-offs** than baseline rendering pipelines and (2) is **complementary** to existing primitive-based pruning techniques. We greatly appreciate your constructive suggestions and believe adding these additional baselines and experiments will help strengthen our work and contributions.
>
> [d] LP-3DGS: Learning to Prune 3D Gaussian Splatting, Arxiv 2024
>
> [e] Mip-Splatting: Alias-free 3D Gaussian Splatting, CVPR 2024
>
> [f] Relaxing Accurate Initialization Constraint for 3D Gaussian Splatting, Arxiv 2024
>
> [i] RadSplat: Radiance Field-Informed Gaussian Splatting for Robust Real-Time Rendering with 900+ FPS, Arxiv 2024
>
> [j] Gaussian Opacity Fields: Efficient High-quality Compact Surface Reconstruction in Unbounded Scenes, Arxiv 2024

---

> ### Author Response · Authors · 2024-08-14
>
> Dear Reviewer ytrw,
>
> Thank you very much again for your constructive review! We believe that including your suggested experiments and clarification will further strengthen our work, which we will incorporate in the final version.
>
> You mentioned that our rebuttal has addressed most of your questions and that you will adjust your evaluation, but we have not yet seen your updated evaluation on our end. As we are approaching the end of the author-reviewer discussion period, we are following up to check whether you need any further information or clarification from our side to assist in your adjustment.
>
> Thank you again for your time and effort in reviewing our paper!

---

### Author Rebuttal · Authors · 2024-08-07

Dear Area Chairs and Reviewers,

We would like to express our gratitude to all the reviewers for their time and effort in providing valuable feedback. Your positive and constructive comments on our paper, particularly regarding its novelty and practical applications, are greatly appreciated. It is gald to see that the concept of fragment pruning and its potential impact has been well-received.

In response to the inquiries requesting additional experiments and further clarifications, we have supplied the requested experiments and provided detailed clarifications, summarized below.

**The following experiments have been provided:**
1. **Error Bars in Latency Profiling:** Included in Fig. 4(b) of the attached PDF.
2. **Extending Fragment Pruning Approach to GS-MCMC:** Results provided in our response to Reviewer ytrw’s W9.
3. **Regularizing the Scale of Gaussian Primitives:** Results included in our response to Reviewer ytrw’s W6 and Fig. 3(a) of the attached PDF.
4. **Benchmarking with Fixed Truncation Threshold:** Results provided in our response to Reviewer RCBw’s W1.
5. **Alternating Optimization Between Truncation Threshold and Other Parameters:** Results included in our response to Reviewer RCBw’s W2.
6. **Additional Visualization of Fragment Density:** Visualization included in Fig. 2 of the attached PDF.
7. **Overfitting Training on All Parameters of 3D Gaussian:** Results included in Fig. 3(c) of the attached PDF.
8. **Quantifying the Training Overhead of Our Proposed Approach:** Results included in the attached PDF.

**Questions Clarified:**
1. **Connection with Two-Stage NeRF to Mesh Framework:** Discussed in our response to Reviewer ytrw’s W4.
2. **Automatic Discovery of the Optimal Threshold and Fragment Density Behavior:** Explained in our response to Reviewer ytrw’s W5.
3. **Choice of Baselines for Benchmark on Static Scenes:** Clarified in our response to Reviewer TxBU’s W1 and Q1.
4. **Choice of Baselines for Benchmark on Dynamic Scenes:** Clarified in our response to Reviewer TxBU’s Q2.
5. **Novelty and Insights of Our Work and Extension to Other Research Areas:** Discussed in our response to Reviewer 7Aqo’s W1.
6. **Justification of Additional Training Costs in Terms of Performance Gains:** Explained in our response to Reviewer 7Aqo’s W2.
7. **Comparison of Reducing Opacity to Zero with Our Proposed Approach:** Clarified in our response to Reviewer 7Aqo’s W3.

We are open to providing further details if any points are still unclear. We would appreciate it if you could review our rebuttal and hope the new experiments and clarifications address your concerns. Please let us know if our responses do not resolve any of your concerns so that we can further clarify.

Best Regards,
Authors of Paper 13442

---

### Decision · Program_Chairs · 2024-09-25

**Decision:**

Accept (poster)

**Comment:**

Following the rebuttal, this paper received all-accept leaning scores of borderline accept, borderline accept, weak accept, weak accept.

The AC concurs with the reviewers and suggests acceptance of this paper as a poster.